# Antioxidants Isolated from *Elaeagnus umbellata* (Thunb.) Protect against Bacterial Infections and Diabetes in Streptozotocin-Induced Diabetic Rat Model

**DOI:** 10.3390/molecules26154464

**Published:** 2021-07-24

**Authors:** Nausheen Nazir, Muhammad Zahoor, Mohammad Nisar, Imran Khan, Riaz Ullah, Amal Alotaibi

**Affiliations:** 1Department of Biochemistry, University of Malakand, Chakdara 18800, Khyber Pakhtunkhwa, Pakistan; nausheen.nazir@uom.edu.pk; 2Department of Botany, University of Malakand, Chakdara 18800, Khyber Pakhtunkhwa, Pakistan; mnshaalpk@yahoo.com; 3Department of Pharmacy, University of Swabi, Anbar 94604, Khyber Pakhtunkhwa, Pakistan; drimran.khan@hotmail.com; 4Department of Pharmacognosy, (MAPPRC), College of Pharmacy, King Saud University, Riyadh 11451, Saudi Arabia; rullah@ksu.edu.sa; 5Basic Science Department, College of Medicine, Princess Nourah bint Abdulrahman University, Riyadh 11671, Saudi Arabia

**Keywords:** morin, oxidative stress, diabetes mellitus, antioxidant enzyme, Gram-positive bacterial strains, Gram-negative bacterial strains, histopathology

## Abstract

The increase in resistance of microbes against conventional drugs is currently a hot issue, whereas diabetes is another main cause of mortalities encountered throughout the world after cancer and heart attacks. New drug sources in the form of plants are investigated to get effective drugs for the mentioned diseases with minimum side effects. *Elaeagnus umbellata* Thunb. is a medicinal plant used for the management of stress related disorders like diabetes and other health complications. The active constituents of the chloroform extract derived from *E. umbellata* berries was isolated by silica gel column chromatography which were identified as morin, phloroglucinol, and 1-hexyl benzene through various spectroscopic techniques (electron ionization mass spectrometry, 1H-NMR, and 13C-NMR spectroscopy). The possible protective effects (antioxidant, antibacterial, and antidiabetic activity) of isolated compounds were evaluated using reported methods. Morin exhibited strong in vitro antiradical potential against DPPH (2,2-diphenyl-1-picrylhydrazyl) and ABTS (2,2′-azinobis-3-ethylbenzothiazoline-6-sulfonic acid) radicals along with prominent antibacterial activities against selected bacterial strains (*Escherichia coli*, *Bacillus cereus*, *Salmonella typhi*, *Klebsiella pneumonia*, *Pseudomonas aeruginosa* and *Proteus mirabilis*). Among the isolated compounds the more potent one (morin) was assessed for its in vivo antidiabetic potential in streptozotocin-induced diabetic rat model. The in vivo effects observed were further confirmed in ex vivo experiments where the effect of isolated compound on antioxidant enzyme like glutathione peroxidase (GPx), total content of reduced glutathione (GSH), % DPPH inhibition, and the lipid peroxidation MDA (Malondialdehyde) level in pancreatic tissues homogenates were evaluated. In vivo morin at tested doses (2, 10, 15, 30 and 50 mg/kg body weight) significantly restored the alterations in the levels of fasting blood glucose level and body weight loss along with significant decrease in levels of cholesterol, triglycerides, low density lipoprotein, HbA1c level, and significantly increased the high-density lipoprotein in diabetic rats. Morin also effectively ameliorated the hepatic enzymes, and renal functions like serum creatinine. Morin significantly increased the antioxidant enzyme like GPx activity, GSH content, and % DPPH inhibition activity, while reduced the lipid peroxidation MDA (malondialdehyde) level in pancreatic tissues homogenates, and modification of histopathological changes in diabetic rats. Morin exhibited high antioxidant, antibacterial, and antidiabetic potentials as compared to phloroglucinol and 1-hexyl benzene, that could, therefore, be considered as a promising therapeutic agent to treat diabetes mellitus and bacterial infections.

## 1. Introduction

During aerobic metabolism, the production of reactive oxygen species (ROS) through stepwise reduction of molecular oxygen in electron-transfer reactions is an unavoidable situation encountered in almost all human bodies. ROS are the prime modulators of cellular dysfunction contributing to disease pathophysiology. The generation of ROS is a continuous physiological process taking place in various cell compartments, including the cytoplasm, cell membrane, endoplasmic reticulum, mitochondria and peroxisome, as part of basal metabolic function and is formed as inevitable by-products of numerous enzymatic reactions. ROS include free radicals such as superoxide anion (O_2_^−^), hydroxyl radical (^•^OH), as well as nonradical molecules like hydrogen peroxide (H_2_O_2_), singlet oxygen (^1^O_2_), and so forth. In the metabolic process, these free radicals act as electron transfer mediators in various biochemical reactions, but excessive production of these free radicals can cause oxidative damage to macromolecules [1]. These free radicals cause structural damage to nucleic acids, proteins, and lipids that is collectively termed as “oxidative stress” [2]. Although biological systems have enzymatic and non-enzymatic antioxidant systems that effectively handle ROS, however, an imbalance of pro-oxidant/antioxidant give rise oxidative stress. Excessive generation of free radicals contributes to the development of various chronic diseases like cancer, diabetes mellitus, cardiovascular diseases, atherosclerosis, rheumatism, ischemic, nephritis, Parkinson’s and Alzheimer’s disease (AD). 

Excessive ROS production can also impair the glucose transporter 4 (GLUT4) and increase the intravascular glucose concentration. GLUT4 is a key component of glucose homeostasis and is responsible for removing glucose from the blood. It has also been reported that ROS is one of the main risk factors for insulin resistance [3]. The pathophysiological processes involved in the etiology of diabetes are insufficient insulin that might be due to insulin resistance or autoimmune damage of islet β-cells. Therefore, inadequate insulin or an altered insulin action in target tissues disrupts the metabolic pathways for carbohydrates, proteins and lipids, which can lead to hyperglycemia [4]. Chronic hyperglycemia results in increased ROS production through increased oxygen consumption, disruption in mitochondrial function, or activation of ROS-producing enzymes. The overproduction of ROS or reduced activity of endogenous antioxidants results in oxidative stress that seems to be due to an imbalance between the oxidizing species as mentioned before, resulting in oxidative stress and cellular death [5].

Clinically, diabetes mellitus (DM) can be characterized by excessive urine excretion, continuous appetite, loss in weight, thirst, fatigue, and blurred vision. The major metabolic disorders associated with DM can lead to keto acidosis, hyper-osmolar coma, neuropathy, renal failure, retinopathy, skin problems, and increasing threats of cardiovascular complications [6]. DM is documented as the seventh cause of death worldwide effecting 100 million people every year. The only way to control T2DM is the use of proper diet, insulin, antidiabetic medications; thiozolidinediones, biguanides, sulfonylureas (glibenclamide, glimepiride), using specific enzyme inhibitors like acarbose and miglitol to reduce demands for insulin. Mostly antidiabetic medications have been isolated from plants sources that reinstate insulin production and hinder the intestinal glucose absorption [7]. 

Researchers have conducted extensive research on the antioxidant properties of various substances, including the natural antioxidant of plant origin that might be effective in preventing complications of diabetes [8]. Antioxidants are bioactive compounds that regulate redox status by blocking or reducing the oxidation of substances like fats, oil, and foods. Numerous studies have shown that antioxidants like lycopene, retinol, α- and γ-tocopherol, β-cryptoxanthin, ascorbic acid, α- and β-carotene, and lutein are present in different plants, essentially reducing diabetes complications [9,10,11]. The tendency in screening the antidiabetic and hypoglycemic potential of medicinal plants has increased, as it provides an important source as they are a good candidate to isolate novel effective drugs for this chronic disease. Due to lack of efficient drugs to treat diabetes, in the meanwhile the World Health Organization (WHO) has recommended healthy food consumption and daily exercise as an effective technique to control type 2 diabetes [12].

Antidiabetic medicinal herbs and their compounds have been confirmed to have beneficial effects in human and also help in relieving complications related with diabetes. Among the medicinal plants *Elaeagnus umbellata* Thunb. has been reported as an antidiabetic medicinal shrub that belongs to the Elaeagnaceae family [13]. It mainly grows in the Himalayan regions of Pakistan and India [13], while it is rare and endangered in Saudi Arabia [14]. The berries extract of *E. umbellata* is a rich source of polyphenols, flavonols, flavones, proanthocyanidins, anthocyanidins, flavonoids, and glycosides with potential antioxidant and enzyme inhibitory activities [15]. *E. umbellata* has great folkloric uses in management of various ailments including the worldwide most prevalent disease of diabetes [13] and neurodegenerative disorder [16] that might be due to various identified phytoconstituents present in it and isolated berries catechins [10]. Polyphenolic compounds like flavonoids are mainly present in berries, vegetables, and green tea which greatly help in curing the oxidative stress related diseases as antidiabetic, anticancer, anti-atherosclerosis, antimicrobial, immunomodulatory, renoprotective, hepatoprotective, and neuroprotective that also improve the human immunity [11]. 

Among the compounds (morin, phloroglucinol, and 1-hexylbenzene) isolated from *E. umbellata* berries antioxidant morin has been assessed for antidiabetic potential. Morin, a bioflavonoid, has demonstrated antioxidant as well as antidiabetic activities along with its antihyperlipidemic effects in streptozotocin-induced diabetic rats [17]. Reported studies have revealed that morin can be used as a non-competitive inhibitor that regulates the expression of MicroRNA-29a (regulator of insulin receptor) in HepG2 cells and improves insulin signaling and stimulating the metabolic pathways [18]. Dietary flavonoid morin has been used in regulating oxidative stress and liver damage induced by high glucose concentration [19].

In recent years, the interest in antioxidants derived from beverages, fruits, herbs, and vegetables has increased. These natural antioxidants can be formulated in the form of nutraceuticals which could be helpful in preventing the oxidative damage. Furthermore, plant antioxidants exhibit its action involving target sites other than those of currently used antibiotics and could be active against drug-resistant microbial pathogens [20]. The overwhelming resistance developed by microbes to available drugs and increase in number of diabetic patients around the globe has compelled scientists around the world to investigate plants being nature factories synthesizing a number of phytochemicals, for potential therapeutic agents. Therefore, these synthetic drugs are constantly replaced by naturally available plant products [21].

Therefore, many plant species have been screened as a valuable source of natural antibacterial agents. Owing to rising incidences of antimicrobial resistance against various chemotherapeutic and antimicrobial agents, the treatment of bacterial infections requires special consideration that may otherwise lead to severe prognosis. Medicinal plants have been a valuable source of medication virtually in all cultures and societies worldwide due to their important antimicrobial phytoconstituents, and wider therapeutic potentials [22]. The bioactive compound isolated from *E. umbellata* has displayed highest potency against both Gram-positive and Gram-negative bacterial strains that have been reported previously [23,24] and now there is need to isolate responsible compounds from it. 

This study aimed to evaluate antioxidant, antibacterial, and antidiabetic potential of compounds isolated from *E. umbellata* berries. The blood glucose level, lipid profile, hepatic and renal functions, antioxidant enzymes, MDA (malondialdehyde) level (lipid peroxidation), and oxidative stress markers in the STZ-induced diabetic rats model were monitored only for isolated morin as it was more potent against the mentioned health complications in in vitro studies. *E. umbellata* was found to be a potent antidiabetic plant that contains a significant number of antioxidants.

## 2. Materials and Methods

### 2.1. Chemicals and Reagents

Antioxidant chemicals such as DPPH, ABTS, and ascorbic acid were purchased from Sigma-Aldrich (St. Louis, MO, USA); while antidiabetic chemicals and reagents such as Type-I α-glucosidase (Baker Yeast), Type-VI α-amylase (porcine pancreas), PNPG (*p*-nitrophenyl-α-D-glucopyranose), acarbose, and streptozotocin were obtained from Sigma Aldrich (Darmstadt, Germany); glucose estimation kits from SD Chek-Gold Germany, and glibenclamide were provided by Sanofi-Aventis-Pharma Pakistan. Chemicals such as Tween-80 (Scharlau-chem., Barcelona, Spain), normal saline solution (Utsoka Pharma, Las Bela Baluchistan, Pakistan), lipid profile tests kits (Human, Hamburg, Germany), renal profile tests kits, and antioxidant enzymes kits (Biomed: Germany; diagnostic) were used in in vivo study. Mueller Hinton (MH) agar and broth (Merck, Modderfontein, South Africa) were used to culture the bacterial strains. Various solvents such as methanol, *n*-hexane, ethyl acetate, and chloroform (Merck, Darmstadt, Germany) were also used in this study.

### 2.2. Plant Material Collection

Berries of *E. umbellata* were collected from local area, in September 2019 and plant specimen was deposited in the Botanical Garden Herbarium, University of Malakand with Voucher No. BGH-UOM-154.

### 2.3. Extraction, Fractionation and Isolation of Morin

*E. umbellata* berries after drying were subjected to extraction and fractionation following procedures in the previously published article [13]. About 10 kg berries were crushed through a mechanical grinder. The final crush fine powders were mixed in 80% methanol for two weeks with periodical shaking and then filtered through muslin cloth followed by Whattman filter paper. Finally, the filtrates were dried into a semisolid mass (750 g) at 45 °C under reduced pressure in a rotary evaporator and then solidified in open air. The resultant methanol extracts were sequentially extracted by solvent–solvent extraction process dissolved in a mixture (0.5 L) of methanol and water (7:3) using different solvents (*n*-hexane, chloroform, ethyl acetate *n*-butanol, and aqueous) starting from a low polarity to high polarity. The chloroform fraction was found the most active on the basis of in vitro test results and was further subjected for isolation of bioactive phytochemical compounds. The chloroform fraction was finally loaded on silica gel chromatography column in solvent system of *n*-hexane and ethyl acetate (3:7) that yielded a pure yellow solid compound-**I** (morin) weighing about 80 mg. A colorless to beige solid compound-II (Benzene-1, 3, 5-triol or Phloroglucinol) was obtained weighing 60 mg (eluted at solvent ration 6:4), while compound-**III** (1-hexylbenzene) was eluted from column in solvent system; *n*-hexane and ethyl acetate (9:1) that yielded 45 mg of **III**. The melting point of isolated compound was 120 °C with Rf. value of 0.47. Different spectroscopic techniques like FTIR, ^1^H NMR, ^13^C NMR, and mass spectrometry were used for the characterization of isolated compound. The purified isolated compound (morin) was screened out for various in vitro antioxidant and antibacterial potential, and in vivo antidiabetic activities. 

### 2.4. Structural Confirmation and Characterization of Isolated Compounds 

The structure formula of morin (2-(2,4-Dihydroxyphenyl)-3,5,7-trihydroxychromen-4-one) is presented in Appendix A. The molecular formula of morin was determined as C_15_H_10_O_7_ by ESIMS and molar mass (*m*/*z*) is 302 g/mol. The FTIR spectra shows aromatic functionality at 1612.49 cm^−1^ while, another broad peak at 3161.33 cm^−1^ demonstrated the C-H bond stretching as presented in Appendix A. ^1^H NMR (Acetone, 300MHz): the ^1^H NMR spectrum is given in Appendix A. The chemical shift from δ = 2 to 2.5 ppm shows the solvent peaks. δ = 12.18 (s, 1H), this singlet peak could be attributed to -OH group proton attached to the carbon number 2 of the said structure. δ = 9.73 (br. s., 1H), this broad peak represents one proton attached to carbon number 22 of the provided structure. δ = 8.58 (br. s., 1H), this singlet peak also represents the proton at position 21 of provided structure. δ = 8.36 (br. s., 1H), this broad peak represents one proton attached on position number 18 of the given structure. δ = 8.01 (s, 1H), this peak could probably represent the proton at position number 12 of given structure. δ = 7.84 (s, 1 H), this peak could be attributed to proton attached at carbon number 9 of the given structure. δ = 7.69 (s, 1H), δ = 6.99 (s, 1H), δ = 6.53 (s, 1H), δ = 6.27 ppm (s, 1H), the last four peaks could be attributed to protons attached at positions 8, 11, 17 and 20. ^13^C NMR (Acetone, 75MHz): δ = 205.6, 175.7, 164.0, 161.4, 156.9, 147.4, 146.1, 144.9, 135.9, 122.9, 120.6, 115.3, 103.3, 98.3, 93.5 ppm (Appendix A). The EIMS of morin is presented in Appendix A. The structure of phloroglucinol is presented in Appendix A while FTIR spectrum in Appendix A. The absorption peaks at 3161.33 and 3479.58 cm^−1^ demonstrate the -OH group. Peak at 2887.44 cm^−1^ indicates C-H stretching. Aromatic nucleus can be recognized by C=C strong peak at 1612.49 cm^−1^. Band at 1739.79 cm^−1^ is due to functionality of 1, 3, 5 substitute benzene (Appendix A). The NMR signals of compound-II in methanol solvent are reported below: 1H NMR (MeOD, 300 MHz) δ = 9.11 (s, 3 H, OH), 5.83 (s, 3 H, C-H) (Appendix A) while 13C NMR (MeOD, 75 MHz) δ160.0 ppm (CH_2_), 95.5 ppm (C-H) (Appendix A). The EIMS is shown in Appendix A. The structure of 1-hexyl benzene is presented in Appendix A. The 1H NMR (DMSO-H2O, 300MHz) spectrum is given in Appendix A. The solvents used were H_2_O and DMSO, the peaks of which are conspicuous in chemical shifts 3.5 and 2.5, respectively. δ = 8.31 (s, 3 H) this singlet most probably represents 3 protons of Benzene ring on position 1, 2 and 3. 7.46 (s, 2 H), this chemical shift could possibly be represented by two protons on positions 4 and 6 of the structure given. δ = 3.17 (t, 2 H, j = 7.2 Hz), these two protons which have been represented by singlet peak could be attributed to position 7 of the structure. δ = 2.09 ppm (m, 08 H), this singlet peak at the highest chemical shift could be attributed to the 08 protons at positions 8, 9, 10 and 11 of the given structure. δ= 0.8 (t, 3H, j = 7.8) present at position 12. 13C NMR (DMSO-d6, 75MHz): δ = 159.6, 148.6, 144.7, 140.1, 138.3, 136.8, 112.8, 112.2, 110.7, 108.1, 79.6, 31.1 ppm (Appendix A). The IR spectrum is presented in Appendix A. The wave number above the fingerprint region represents the carbon–carbon double bond of aryl group from 1616 to 1691 cm^−1^. The region from 3000 to 3100 cm^−1^ could be attributed to the C-H bond with SP_2_ hybridization. The EIMS is presented in Appendix A.

### 2.5. DPPH (2,2-Diphenyl-1-picrylhydrazyl) Free Radical Scavenging Test Assay

According to the Brand-Williams assay [25] DPPH (2, 2-diphenyl-1-picrylhydrazyl) free radical scavenging activity of morin was determined. DPPH and compounds stock solutions (1 mg/mL) were prepared in 100 mL methanol with serial dilutions of 31.05, 62.5, 250, 500, and 1000 µg/mL. After that, 0.1 mL morin was mixed with 3.0 mL DPPH solution and incubated at 23 °C for 30 min. Standard ascorbic acid solution was also prepared in the same way using as a positive control. Finally, absorbance was measured at 517 nm via UV spectrophotometer. Results were taken in a set of three experiments and represented as mean ± SEM. Percent free radicals scavenging potential was calculated according to the following Equation (1): (1)$$\%\;\mathrm{Free}\;\mathrm{radicals}\;\mathrm{Scavenging}\;\mathrm{potential}\;=\frac{\;\mathrm{Blank}\;\mathrm{sample}\;\mathrm{absorbance}-\mathrm{sample}\;\mathrm{absorbance}}{\mathrm{Blank}\;\mathrm{sample}\;\mathrm{absorbance}}\times 100$$

### 2.6. ABTS (2,2′-Azinobis-3-ethylbenzothiazoline-6-sulfonic acid) Free Radical Scavenging Test Assay 

ABTS (2,2′-azinobis-3-ethylbenzothiazoline-6-sulfonic acid) free radicals inhibition potential were determined according to the reported assay [26]. ABTS solution was prepared in methanol (100 mL). In total, 300 µL of compound (morin, phloroglucinol, and 1-hexyl benzene) was mixed thoroughly with 3.0 mL ABTS solution and incubated for six minutes. Finally, the absorbance was measured through UV spectrophotometer using ascorbic acid as positive control. Percent ABTS free radicals scavenging potential was calculated using the equation No. 1.

### 2.7. In Vitro α-Amylase Inhibition Assay

The α-amylase inhibition activity of morin was assessed using 3, 5-dinitrosalicylic acid (DNSA) assay [27]. Morin, phloroglucinol, and 1-hexyl benzene were dissolved in mixture of 10% DMSO, 0.02M Na_2_HPO_4_/NaH_2_PO_4_ buffer, and 0.006M NaCl (pH 6.9) and finally dilutions (31.05, 62.5, 125, 250, 500 and 1000 µg/mL) were prepared. In total, 200 μL α-amylase (2 units/mL) was mixed with 200 μL morin and incubated at 30 °C for 10 min. After that, 200 μL starch (1%) was added to each serial dilution and incubated for 3 min. To stop the reaction 200 μL sodium potassium tartrate tetrahydrate reagent, 2M NaOH, and 20 mL DNSA (96mM) reagent were added to the reaction mixture. The reaction mixtures were boiled for 10 min at 85–90 °C in a water bath. After cooling the mixture was diluted with 5mL distilled water and finally at 540 nm absorbance was recorded. A blank solution was prepared containing only isolated compound morin but no enzyme. Standard acarbose (100 μg/mL–2 μg/mL) was prepared as positive control. The α-amylase enzyme inhibitory activity was calculated by the following Equation (2): (2)$$\%\;\alpha \;-\;\mathrm{amylase}\;\mathrm{Inhibition}\;=\frac{\mathrm{control}\;\mathrm{absorbance}-\mathrm{sample}\;\mathrm{absorbance}}{\mathrm{control}\;\mathrm{absorbance}}\times 100$$

### 2.8. In Vitro α-Glucosidase Inhibition Assay 

The α-glucosidase inhibition potential of morin was determined according to the reported method [27] with minor changes. The reaction mixture was prepared by taking 100 μL α-glucosidase (0.5 unit/mL), 600 μL phosphate buffer (0.1 M; pH 6.9), and 50 μL of isolated compounds (morin, phloroglucinol, and 1-hexyl benzene) with serial dilutions of 31.05, 62.5, 125, 250, 500 and 1000 µg/mL. All the reaction mixtures were incubated at 37 °C for 15 min. The enzymatic reactions were started by adding 100 μL substrate (*p*-nitro-phenyl-α-D-glucopyranoside) solution (5 mM) formed in phosphate buffer (0.1 M; pH 6.9), finally incubated at 37 °C for 15 min. The reaction was completed by adding sodium carbonate (400 μL; 0.2 M) solution and then at 405 nm absorbance was measured. The blank solution was prepared containing only isolated compounds without α-glucosidase. Percent α-glucosidase inhibition potential was calculated using Equation (3): (3)$$\%\;\alpha \;-\;\mathrm{glucosidase}\;\mathrm{Inhibition}\;=\frac{\mathrm{control}\;\mathrm{absorbance}-\mathrm{sample}\;\mathrm{absorbance}}{\mathrm{control}\;\mathrm{absorbance}}\times 100$$

### 2.9. Evaluation of the Antimicrobial Activity

#### 2.9.1. Bacterial Strains

Antibacterial potential of isolated compound was investigated against both Gram-positive and Gram-negative bacterial strains; *Escherichia coli*, *Bacillus cereus*, *Salmonella typhi*, *Klebsiella pneumonia*, *Pseudomonas aeruginosa* and *Proteus mirabilis*. Bacterial strains were provided by the Department of Pharmacy, University of Malakand, Khyber Pakhtunkhwa, Pakistan, and were kept at 4 °C in agar slants in freeze-dried condition until further use.

#### 2.9.2. Preparation of Standard Bacterial Suspension

Both Gram-positive and Gram-negative bacterial strains *Escherichia coli*, *Bacillus cereus*, *Salmonella typhi*, *Klebsiella pneumonia*, *Pseudomonas aeruginosa* and *Proteus mirabilis* organism viable count per mL in the stock suspension was determined using surface viable counting technique. About 10^8^–10^9^ colony-forming units (CFU) per mL were used as inoculum in subsequent experiments. A fresh stock suspension was used each time.

#### 2.9.3. Antibacterial Susceptibility Test (Disc Diffusion Method)

The antibacterial activity of the isolated compounds (morin, phloroglucinol, and 1-hexyl benzene) was determined using agar well diffusion method [28]. Muller–Hinton agar media was prepared and autoclaved at 15 lbs pressure for 20 min and cooled to 45 °C. The standard bacterial stock suspension 10^8^–10^9^ CFU/mL were spread on Muller–Hinton agar plates and five wells (3 mm in diameter; four on side and one in center) were punched in plates using a sterile cork borer. The entire wells were filled with 100 µL of the isolated compounds (stock solution: 1 mg/mL) using microtiter-pipette and allowed to diffuse at room temperature for 2 h. The plates were incubated at 37 °C for 24 h. In the center hole, Imipenem was added which served as positive control. Simultaneous, addition of an equal concentration of DMSO solvent instead of isolated compounds was used as blank. After incubation, the zones of inhibition were measured in millimeters (mm) and mean values were calculated. The experiments were performed in triplicates.

#### 2.9.4. Determination of Minimum Inhibitory Concentration of Isolated Compounds

Minimum inhibitory concentrations (MICs) of morin, phloroglucinol, and 1-hexyl benzene were determined using broth dilution method [29], following the Clinical and Laboratory Standards Institute (CLSI) guidelines. Stock solutions of morin, phloroglucinol, and 1-hexyl benzene were prepared in dimethylsulphoxide (DMSO). The serial dilutions from the stock solution were made ranging from 500 to 3.90 μg/mL using Mueller–Hinton broth. Sterile centrifuge tubes were arranged in sets of rows for each bacterial inoculum to cover the different concentration of isolated compounds in triplicate. The standard bacterial stock suspension 10^8^–10^9^ CFU/mL, in 100 μL were inoculated into each tube. A sterility control tube and a growth control tube were also incubated (for each strain). The tubes were incubated at 37 °C for 24 h. After incubation, 40 μL of a 0.4 mg/mL solution of para iodonitrotetrazolium (INT) chloride was added to each tube as an indicator of microbial growth. All the tubes were incubated at 37 °C for 30 min and the MIC values were visually determined. The evaluation for the inhibition of microbial growth was based on the color change of INT to pink, while no color change indicates the absence of growth. The MIC is the lowest concentration of the compounds at which there is no visible growth in the tubes. The concentration that inhibited bacterial growth completely (the first clear tube) was taken as the MIC value. Experiments were performed in triplicate and mean values are presented.

In order to, determine the sensitivity of the bacterial strains, positive control experiments were conducted, taking ampicillin at a starting concentration of 0.10 mg/mL in sterile water while the final concentrations for these experiments ranged from 25.00 to 0.19 μg/mL. A negative control experiment was conducted using only DMSO. 

### 2.10. In Vivo Studies

#### 2.10.1. Animals

Sprague Dawley adult albino rats (body weight range = 170–200 g) were used in in vivo studies. The animals were purchased from Rifah Institute of Pharmaceutical Sciences Islamabad. The entire animals were given food and fresh water ad libitum. The animals were adjusted at room temperature around 22–25 °C. The animals were housed three rats per cage and maintained with light/dark cycle of about 12 h. All the in vivo procedures were carried out according to the ARRIVE guidelines and approval was taken from the Departmental Animal Ethical Committee (DAEC/2019/1), University of Malakand. 

#### 2.10.2. Study design

All the animals were separated into nine groups having eight animals each (Table 1). In the current investigational study STZ-induced diabetic model was used to evaluate the anti-hyperglycemic and anti-hyperlipidemic activity of isolated compound morin in chronic multiple dose experiment. The schematic treatment groups, given dose and route of administration are presented in Table 1. All the animals were treated with their respective treatments for 21 days starting daily at 09:00 am. The fasting blood glucose levels were determined at 0, 4th, 7th, 10th, 15th, and 21st days with glucometer from the tail vein. Changes in the body weight were recorded for untreated control, diabetic, and treated groups in fasting condition from day 1 to day 21 of experimental period and are presented as % increase in body weight [13].

#### 2.10.3. Acute Toxicity Study 

For acute toxicity study all the animals were treated orally with different doses (50, 100, 150, 200, 250, 300, and 500 mg/kg body weight) of isolated compound morin. After that the animals were kept under observation for 4 h for signs of toxicity (respiratory aches, convulsions, motor activity, tremors, muscle spasm, sedation, diarrhea, loss of righting reflex, hypnosis, and salivation). All the animals appeared healthy for 24 h to 1 week with no noticeable changes in appearance or behavior, and no signs of mortality were noticed. Morin was found safe and nontoxic up to 500 mg/kg body weight dose range. Therefore, according to the Organization for Economic Cooperation and Development (OECD) and ARRIVE (Animal Research: Reporting of In Vivo Experiments) guidelines [30], morin at dose of 50 mg/kg body weight that is 1/10th of the highest dose (500 mg/Kg) was selected for evaluation of antidiabetic activity. 

#### 2.10.4. Induction of Diabetes in Experimental Animals

According to the reported protocol [31] diabetes was induced into *Sprague Dawley* adult albino rats by intraperitoneal injection of STZ (50 mg/kg body weight). Blood samples were collected from the tail vein through glucometer strips by means of SD glucometer and the level of blood glucose was measured 72 h after STZ injection. Normal control groups were given normal saline (8 mL/kg, p.o.) as a vehicle, while isolated compound morin to the treatment groups from 3 days after STZ administration for 3 weeks. Animals with blood glucose levels higher than 300 mg/dL were considered diabetic. During the experiment the blood glucose level and body weights were measured. 

#### 2.10.5. Collection of Blood and Estimation of Biochemical Parameters 

On 21st day of in vivo antidiabetic activity, all the animals were anesthetized by intraperitoneal (IP) injection of pentobarbital sodium (35 mg/kg body weight) and euthanized by means of cervical decapitation according to the ARRIVE guidelines. Blood samples were collected by cardiac puncture and transferred to EDTA containing tubes. Blood was centrifuged at 3500 rpm for 10 min for the separation of serum. Biochemical parameters like serum alkaline phosphatase (ALP), serum glutamate pyruvate transaminase (SGPT), serum glutamate oxaloacetate transaminase (SGOT) and lipid parameters like total cholesterol (TC), triglycerides (TG), low-density lipoproteins (LDL), high-density lipoprotein (HDL), and serum creatinine in serum were assessed by using Biochemistry analyzer (PS-520; Shenzhen Procan Electronics-China) using diagnostic kits (Reactivos, GPL Barcelona, Spain) [32]. According to the reported protocol an enzyme-linked immunosorbent assay (ELISA) kit was used to evaluate the level of glycosylated hemoglobin (HbA1c) [33]. 

#### 2.10.6. Measurement of Antioxidant Enzymes Activities and Oxidative Stress Markers in Pancreas Tissues

The pancreas tissue (1 g) was isolated from Sprague Dawley rats, washed with cold saline solution and blotted dried. A weighed portion of the pancreas tissue was homogenized on ice and then centrifuged for 15 min at 3500 rpm at 4 °C. The supernatant was then taken and stored at −20 °C until assessed for different enzymes. 

The activities of reduced glutathione (GSH) level were determined in the pancreatic tissues according to the previously established method [34]. About 300 µL of tissue supernatant was centrifuged for 5 min at 3500 rpm following deproteinization with 300 µL of 10% trichloroacetic acid. In total, 200 µL of the resulting supernatant of each sample (at concentrations 2, 5, 10, 15, 30, and 50 µg/mL) was mixed with 50 µL of Ellman’s reagent (5,5′-dithiobis-(2-nitro-benzoic acid)) (DTNB) then centrifuged at 3000 rpm for 15 min. The absorbance was recorded at 412 nm. A series of standards treated in a similar way were also run to determine the glutathione content. The amount of glutathione is expressed as µmole/mg of tissue protein.

Lipid peroxidation MDA level of the tissue was determined according to a previously published thiobarbituric acid reactive substance (TBARS) assay [35]. MDA, an end product of lipid peroxidation reacts with thiobarbituric acid (TBA) to form a colored substance. Measurement of MDA by thiobarbituric acid reactivity is the most widely used method for assessing lipid peroxidation. The pancreas tissue homogenates were reacted with deproteinizing reagent containing 10% (*w*/*v*) trichloroacetic acid and 50 mg/L (*w*/*v*) butylated hydroxytoluene (BHT). After centrifugation, 200 µL tissue supernatant obtained from each experimental animal was taken and incubated with the chromogenic solution containing 0.44  M H_3_PO_4_ and 0.6% (*w*/*v*) TBA at 90 °C for 30 min. The pink colored TBARS was measured photometrically at 532 nm and tissue MDA was determined from a standard curve and is expressed as nmole/mg of tissue protein.

The activity of glutathione peroxidase (GPx) level was assayed according to the reported method [36]. The reaction mixture contained 0.2 mL of EDTA, 0.1 mL of sodium azide, 0.1 mL of H_2_O_2_, 0.2 mL of reduced glutathione, 0.4 mL of phosphate buffer and 0.2 mL tissue homogenate that were incubated at 37 °C for 10 min. The reaction was stopped by addition of 0.5 mL of TCA, and the tubes were centrifuged at 2000 rpm. To the supernatant, 3 mL of disodium hydrogen phosphate and 1.0 mL DTNB were added, that resulted in color development and the enzyme activity was measured at 420 nm immediately. 

#### 2.10.7. Quantification of Intracellular ROS Level in Pancreas Tissue

Cellular oxidative stress due to the overproduction of reactive oxygen species (ROS) generated by streptozotocin was measured in pancreas tissues through DPPH free radical scavenging assays [25]. Briefly, 50 µL (0.3 mM) DPPH in methanol was incubated with 100 µL supernatant of the tissue homogenates of each experimental group and animals were kept in the dark for 30 min at room temperature. Ascorbic acid was used as the standard. The absorbance was then read against a blank at 517 nm. The percentage scavenging was calculated according to the equation No 1.

#### 2.10.8. Histopathological Assessment

For histopathological assessment, the pancreas tissues were removed immediately and washed with cold saline solution to remove the blood. Finally, the pancreas portions were fixed in 10% formalin, dehydrated with mixture of ethanol/xylene and then fixed with paraffin. Staining of tissue slides was done with hematoxylin and eosin (H & E) dye through automatic slide stainer (Sakura Tissue-Tek^®^ DRS^™^ 2000; Tokyo, Japan). According to the reported assay [37], the stained slides were observed under microscope to see any variation in pancreatic tissue architecture of STZ-induced diabetic group and treated (glibenclamide/morin) groups.

### 2.11. Statistical Analysis 

All the in vitro and in vivo experiments were analyzed in three replicates and determined as mean ± SEM. Student’s t-test and one way ANOVA followed by Dunnett’s post hoc multiple comparison test was used. *p* ≤ 0.05 were considered as significant. Linear regression analysis was used to calculate the IC_50_ value for % DPPH, ABTS, α-amylase, and α-glucosidase inhibition against the different concentrations of the test samples by means of Excel program 2007. 

## 3. Results

### 3.1. In Vitro Free Radical Scavenging Potential of Isolated Compounds (Morin, Phloroglucinol, and 1-Hexyl Benzene)

Morin showed highest percent free radical scavenging activity against DPPH (88± 1.1) and ABTS (88 ± 1.0) with lowest IC_50_ values of 37 and 40 µg/mL, while phloroglucinol and 1-hexyl benzene showed lowest percent free radical scavenging activity against DPPH and ABTS at maximum concentration of 1000 µg/mL with IC_50_ highest values of 105 and 112 µg/mL, respectively (Table 2). The results were comparable with standard ascorbic acid which exhibited highest % antiradical potential (DPPH: 86 ± 0.5; ABTS: 85 ± 0.5) 1000 µg/mL concentration with IC_50_ values 29 and 32 µg/mL.

### 3.2. In Vitro α-Amylase Enzyme Inhibitory Studies of Isolated Compounds (Morin, Phloroglucinol, 1-Hexyl Benzene) 

The IC_50_ values were calculated by evaluating the plot of % α-amylase enzyme inhibition as a function of isolated compound (morin, phloroglucinol, 1-hexyl benzene) concentrations (Table 3). % α-amylase inhibition potential of morin was 75 ± 1.5 *** with IC_50_ values 55 µg/mL at highest concentration 1000 µg/mL, respectively. Phloroglucinol and 1-hexyl benzene showed lowest % α-amylase inhibition potential. Acarbose was used as a standard which causes 90 ± 0.5 percent inhibition at maximum concentration of 1000 µg/mL with IC_50_ value 25 µg/mL.

### 3.3. In Vitro α-Glucosidase Enzyme Inhibitory Studies of Isolated Compounds (Morin, Phloroglucinol, 1-Hexyl Benzene)

Percent α-glucosidase inhibition potential of morin was 79 ± 1.0 *** with IC_50_ value 40 µg/mL at highest concentration 1000 µg/mL, respectively. The IC_50_ value was calculated from the serial dilutions (31.05, 62.5, 250, 500, and 1000 µg/mL) plot of % α-glucosidase inhibition as a function of isolated compounds (morin, phloroglucinol, 1-hexyl benzene) concentrations (Table 3). Phloroglucinol and 1-hexyl benzene exhibited lowest % α-glucosidase inhibition potential. The results were compared with standard acarbose which causes 89 ± 0.5 inhibition at maximum concentration 1000 µg/mL with IC_50_ value 28 µg/mL.

### 3.4. Determination of Zone of Inhibitions 

The antibacterial activity of isolated compounds morin, phloroglucinol, and 1-hexyl benzene was studied in concentration range of 100 µg/mL. Antibacterial potential was assessed in terms of zone of inhibition of bacteria growth due to presence of isolated compounds that inhibits their growth. Morin exhibited maximum zonal inhibition against all the Gram-positive bacterial strains; *E. coli*, *B. cereus*, *S. typhi*, *K. pneumnonia* and *P. aeruginosa* and Gram-negative bacteria; *P. mirabilis* and *S. aureus*. Isolated compound (morin, phloroglucinol, and 1-hexyl benzene) exhibited the inhibitory zone against the tested strains like *E. coli* (19 ± 1.1 **, 15 ± 0.3 *** and 9.3 ± 1.1 *** mm), *B. cereus* (17 ± 0.1 **, 13 ± 1.2 *** and 8.5 ± 0.1 *** mm), *S. typhi* (16 ± 1.2 ***, 12 ± 1.1 *** and 7.4 ± 1.2 *** mm), *K. pneumnonia* (15 ± 0.5 ***, 11 ± 0.5 *** and 6.3 ± 0.5 *** mm), and *P. aeruginosa* (14 ± 0.5 ***, 10 ± 1.1 *** and 5.5 ± 0.5 *** mm) at 100 µg/mL concentration. While. zone size inhibited by morin, phloroglucinol, and 1-hexyl benzene for Gram-negative bacteria; *P. mirabilis* was measured 11 ± 1.5 ***, 7.5 ± 0.2 ***, and 4.5 ± 1.5 *** mm and for *S. aureus* was 13 ± 0.4 ***, 9.5 ± 2.1 *** and 3.4 ± 0.4 *** mm at 100 µg/mL concentration. Results indicated that morin exhibited highest inhibitory zone against both Gram-positive and Gram-negative bacterial strains, the growth inhibition zone measured ranged from 13 to 19 mm. The results showed morin was found to be more effective against bacterial strains used as compared to phloroglucinol and 1-hexyl benzene and hence the plant species was selected for further isolation of antibacterial compounds. The antibacterial effect of isolated compounds is presented in Table 4 which was almost comparable to the effects of the positive control. Imipenem, a broad spectrum drug, was used as standard with noticeable zonal inhibition against the tested bacterial strains. The following tendencies of bacterial strains sensitivity to compounds were observed as *Escherichia coli > Bacillus cereus > Salmonella typhi > Klebsiella pneumonia > Pseudomonas aeruginosa > Staphylococcus aureus > Proteus mirabilis* (Table 4).

### 3.5. Determination of Minimum Inhibitory Concentrations (MICs) 

The antibacterial activities of the isolated compounds are shown in Table 5. All bacterial strains were sensitive against all tested compounds. Results indicated that antibacterial properties of morin were more pronounced against Gram-positive bacterial strains rather than Gram-negative bacterial strains with lowest MIC values ranging from 21 ± 1.2 to 68 ± 1.3 μg/mL. The minimum concentration of morin was evaluated using agar dilution method. Phloroglucinol and 1-Hexyl benzene showed high MIC values as compared to morin.

### 3.6. In Vivo Antidiabetic Potential of Morin

#### 3.6.1. Effect of Morin on Blood Glycemia 

The effect of morin and standard glibenclamide on blood glucose level in normal control, diabetic control, and compound treated groups are shown in Figure 1. Oral administration of the compound (30, 15, 50 mg/kg body weight) caused a significant (*p* ˂ 0.01) decrease in blood glucose level as compared to diabetic control. Blood glucose level reduction was observable from the 7th day and onward. The compound at doses 2 and 5 mg/kg body weight slowly reduce blood glucose level while compounds at 10, 15, 30, and 50 mg/kg body weight doses displayed comparable decline in glucose level compared to the diabetic control group. Standard drug glibenclamide (5 mg/kg, body weight) significantly (*p* < 0.001) decreased the fasting glucose level as compared to diabetic control rats on day 21 of experimental period. Furthermore, blood glucose level reduction was apparent from the 7th day and onward, while the highest decline was observed on day 21 (*p* < 0.01; *p* < 0.001) in the morin treatment groups.

#### 3.6.2. Effects of Morin on Body Weight in Diabetic Rats

The effect of morin (2, 5, 10, 15, 30 and 50 mg/kg body weight) and standard glibenclamide on changes in body weight in normal control group, diabetic control, and compound (morin) treatment group are shown in Figure 2. STZ-induced diabetic rats revealed significant (*p* ˂ 0.001) reduction in body weight (128.23 ± 2.84) as compared to normal control rats. Diabetic rats treated with glibenclamide (0.5 mg/kg, p.o.) had significantly (*p* < 0.001) reversed the diabetes-induced decrease in body weight as compared to diabetic control rats. Diabetes is characterized by loss in body weight and thus was observed in this study. However, the decrease in body weight of the rats was improved by the treatment of the isolated compound morin and glibenclamide. Morin at highest doses, 15, 30, and 50 mg/kg body weight showed significant (*p* < 0.001) improvement in body weight gain which was comparable to standard drug glibenclamide. While, the selected compound at dose 10 mg/kg body weight also showed a significant (*p* < 0.01) increases in body weight gain.

#### 3.6.3. Effect of Various Treatments on Liver and Renal Functions in STZ-Induced Diabetic Rats

The activity of hepatic enzymes like ALP (alkaline phosphatase), SGPT (serum glutamate pyruvate transaminase), SGOT (serum glutamate oxaloacetate transaminase), and renal functions like serum creatinine in normal control group, diabetic control group, and morin treated group are shown in Figure 3. A considerable increase (*p* < 0.01; *p* < 0.001) occurs in the level of liver marker enzymes like ALP (Figure 3A), SGPT (Figure 3B), and SGOT (Figure 3C) and renal functions like serum creatinine (Figure 3D) in diabetic control group as compared to the normal control. However, administration of standard glibenclamide (5 mg/kg) and morin (15, 30, and 50 mg/kg body weight) to diabetic rats significantly reduced (*p* < 0.01; *p* < 0.001) the levels of ALP, SGPT, SGOT, and serum creatinine.

#### 3.6.4. Effect of Various Treatments on Serum Lipid Profile and Hb1c Level in Diabetic Rats 

Diabetic control group rats showed significant (*p* < 0.01; *p* < 0.001) increases in plasma lipid parameters like TC (Figure 4A), TG (Figure 4B), LDL (Figure 4D), and HbA1c levels (Figure 4E), but a significant (*p* < 0.01) decline was observed in plasma HDL level (Figure 4C) as compared to normal control group rats. Standard glibenclamide (0.5 mg/kg) and morin at highest doses 15, 30, and 50 mg/kg has shown a significant decrease (*p* < 0.01; *p* < 0.001) in the levels of lipid profile; TC, TG, LDL, and HbA1c levels, while significantly increased (*p* < 0.01) HDL cholesterol in diabetic rats as compared to diabetic control rats.

#### 3.6.5. Effect of Various Treatments on Lipid Peroxidation and Reduced GSH Level

Figure 5 shows that administration of morin (2, 5, 10, 15, 30, and 50 µg/kg body weight) significantly (*p* < 0.05, *p* < 0.01, *p* < 0.001) decreased MDA level in STZ-induced diabetic rats. The high dose of morin (50 µg/kg) was more effective as compared to low doses (10 mg/kg and below). Morin treated groups exhibited a significant decrease of GSH level estimated in pancreas tissue homogenates in comparison to diabetic control group (Figure 5A). Furthermore, a significant reduction in lipid peroxidation, observed through significant decrease in the concentration of MDA level estimated in pancreas tissue homogenates of the morin treated groups in comparison to diabetic control group, was also observed (Figure 5B).

#### 3.6.6. Effect of Various Treatments on Glutathione Peroxidase Level

Biochemical analysis showed that antioxidant stress marker enzyme levels like glutathione peroxidase in pancreas tissue homogenates of STZ-induced diabetic rats were significantly reduced which were controlled by treatment with morin isolated from berries of *E. umbellata* in STZ-induced diabetic rats suggesting that the plant possesses strong antioxidant proprieties revealed from ex vivo activities (Figure 5C).

#### 3.6.7. Effect of Various Treatments on Intracellular ROS Level

Ex vivo percent DPPH free radicals scavenging potential of pancreas tissue of different animal groups are summarized in Figure 5D. Percent DPPH inhibition activity in the pancreas of normal control group (75.55 ± 3.12) was significantly (### *p* < 0.001) high as compared to diabetic control group (22.41 ± 4.25). % DPPH inhibition activity of glibenclamide and morin (2, 5, 10, 15, 30, and 50 µg/kg body weight) treated groups in the pancreas tissues were 49.88 ± 3.21 **, 52.66 ± 2.81 **, 60.42 ± 2.12 ***, 65.33 ± 22.13 ***, 69.33 ± 3.12 ***, and 71.42 ± 3.23 *** which were significantly higher than that of diabetic control group.

#### 3.6.8. Effect of Various Treatments on Pancreas Histopathology in STZ-Induced Diabetic Rats 

The effect of morin in different doses on pancreas histopathology in STZ-induced diabetic rats is represented in Figure 6. The pancreas of the control group rats showed a normal, clear lobular histological architecture, the cells of the islets of Langerhans had normal morphologies, and the acinar cells were clear with prominent and centrally placed nuclei and interlobular spaces (Figure 6A). Section of pancreas from diabetic control group of rats showed a typical histological pattern with slight grade of congestion and dilated blood vessels with inflammatory cell infiltration, reduction in the size of pancreatic islets with decreased cellular density. The boundary of endocrine and exocrine portion of pancreas was also indistinct (Figure 6B). Histopathology of a pancreas from standard glibenclamide (5 mg/kg/overly) treated group did not show any violation from the normal histological pattern during the experimental period. No congestion or deterioration was observed in the in acinar cells. There was no reduction in the diameter of Islet of Langerhans and nuclei were also centrally placed. The boundary of endocrine and exocrine portion of pancreas was distinct and overall normal histological pattern was retained (Figure 6C). Histological pattern of morin (2, 5 mg/kg/orally) treated group of rats did not show restoration of pancreatic tissue. Pancreatic histology revealed a decrease in the size of pancreatic islet cells, nuclei were not properly aligned, and an increase in inflammatory cells was also observed to some extent (Figure 6D,E). The rats treated with morin (10, 15 mg/kg/orally) showed significant dose dependent improvement in cellular architecture as observed by the restoration of the normal cellular population size of islets cells, with minimal reduction in the size of pancreatic islets, a recovery from necrosis in the pancreatic acini, and the overall cytoplasmic integrity was retained to some extent (Figure 6F,G). Improved architecture of pancreas is observed in groups treated with morin (30, 50 mg/kg/orally) is showing normal histological pattern of pancreas with clear lobular architecture, Islet of Langerhans was of normal diameter and structure, acinar cells were clear with prominent and centrally placed nuclei and interlobular spaces were also visible. Moreover, the central cellular integrity and lobular structure of pancreas was retained (Figure 6H,I). This result confirmed that morin at dose of 30 mg/kg/orally and 50 mg/kg/orally showed maximum effectiveness for treatment of STZ-induced diabetes and pancreatic morphologies resembling those of normal rats.

## 4. Discussion

In recent years, medicinal plants have seemed as potential sources of antioxidants, antidiabetic, antimicrobials, and secondary metabolites for therapeutic interventions, which have opened doors for the development of novel plant-based antibacterial agents [38,39]. However, the problems associated with these antimicrobial drugs are their toxic effects and the resistance shown by microbes against these specific groups of antimicrobial agents. Therefore, these synthetic drugs have been replaced by naturally available plant products [22,40]. The current study was designed on the isolated compound (morin) by keeping in view the reported antibacterial action of *E. umbellata* plant extract [23,24]. Morin exhibited relatively higher antibacterial activity against Gram-positive than Gram-negative bacterial strains. The reason for higher sensitivity of the Gram-positive bacterial strains than Gram-negative bacteria could be ascribed to their variations in cell membrane constituents. Gram-positive bacteria contain an outer peptidoglycan layer, which is an ineffective permeability barrier. Our results were in line with the previously reported study [34]. The reported antioxidant and antibacterial activity of *E. umbellata* extracts might be due to the high content of flavonoids [23,24]. The results indicated broad spectrum action of isolated flavonoid morin. The findings of this study specify that the plant *E. umbellata* can potentially be used against infectious diseases. 

The simplest product of carbohydrate metabolism is glucose which is readily absorbed into the blood and caused hyperglycemia. Post-prandial hyperglycemic condition can also be developed by inhibiting carbohydrate digesting enzymes; α-amylase and α-glucosidase. In our results, isolated berries flavonoid morin exhibited strong inhibition of carbohydrate digestive enzymes. Our results were similar to the reported studies on plant extract/fraction that showed strong inhibitory action on both α-amylase and α-glucosidase [13,39,41]. On the basis of our previous reported study, *E. umbellata* could be considered a potential antidiabetic medicinal plant and its berry phytoconstituents and glibenclamide can be good therapeutic agents that sufficiently inhibit the rise in glucose level in streptozotocin induced diabetic rat model by inhibiting the activity of carbohydrate digestive enzymes and minimize the post prandial hyperglycemia [13].

The results of current study showed that morin at the dose range of 30–50 mg/kg body weight showed significant improvement in glucose level, lipid profiles (TC, TGs, LDL, and HDL), liver functions (ALP, SGPT, SGOT, and serum creatinine), and body weight in the serum of animals treated with streptozotocin. Reported literatures such as Lo et al. [42] and Dzydzan et al. [43] revealed that treatment with STZ causes an imbalance between plasma oxidant and antioxidant content causes oxidative stress and accelerates the progression of T2DM and its related complications. Hyperglycemia caused due to T2DM also linked with the generation of oxygen free radicals that leads to oxidative stress. Streptozotocin reduces insulin secretion after entering to the pancreatic β-cell through the low-affinity of glucose protein-2 transporter and causes the selective damage of the insulin-producing islet β-cells [13,44,45].

Phenolic compounds like flavonoids have shown strong antioxidant activity and reported as best scavenger of free radicals. Morin also exhibited its free radicals scavenging ability in STZ-diabetic rat model. A significant increase was observed in the lipid parameters like TC, TGs, and LDL cholesterol, HbA1c level, and liver parameters like ALP, SGPT, SGOT, serum creatinine, while HDL cholesterol level was significantly decreased in STZ-induced diabetes in rats as compared to the normal control group animal. In this study, a key biomarker of diabetes Hb1c level significantly increased in STZ-induced diabetic rat model which has also been reported in previous studies [46]. The volume of glucose that enters the erythrocyte membrane has greatly affected the HbA1c level, either by increasing the glycosylation rate or changing the lifetime of erythrocytes that might result in an increased or decreased HbA1c value, respectively [46]. Some reported studies were also in line with the current study that elevation in TGs and LDL or apo lipoprotein-B contents, and low HDL cholesterol level are due to high level of circulating free fatty acids which strongly linked with the development of an atherogenic dyslipidemia [47]. In the current study, lipid and liver parameters were significantly decreased in morin treated diabetic rats compared to the untreated diabetic rat model that designated its protective activity against STZ induced effects. Reported literature [13,48] by other researchers also confirmed the antioxidant, hypoglycemic, hypolipidemic, and reduced lipid peroxidative effects using plant extracts in the STZ-induced diabetic rat. 

Loss in body weight was observed after 21 days treatment in STZ-induced diabetic rats due to excessive tissue protein breaking down. A decrease in body weight is strongly linked with reduction in carbohydrate reserve as an instant source of energy and fats catabolism. In the current study, it was reported that oral administration of morin at dose range of 30–50 mg/kg body weight showed a marked increase in body weight that was in line with the previously reported studies [10,13,49] in a diabetic rat model.

Histopathological study results also revealed an improvement in morin treated animal groups as compared to STZ-induced diabetic group. Pancreatic islet cells of Langerhans are showing degeneration and vacuolization that leads to reduction in cellular density. The borderline of pancreas endocrine and exocrine portion was unclear. According to the reported literature [50], a decrease in islets β-cells number and size was also observed. Normal histopathological pattern of pancreas was observed for glibenclamide (50 g/kg/overly) and morin (15, 30, 50 mg/kg; p.o.) treated groups that strongly correlate with previously published data [51]. The potential effect of morin on streptozotocin-induced diabetic rats is presented in Figure 7. Moreover, based on the antioxidant, antibacterial, and in vitro and in vivo antidiabetic potential and biochemical results of isolated compound morin, we hypothesize that *E. umbellata* could possibly act directly as a free radical scavenger or regulator to inhibit carbohydrate digesting enzymes; α-amylase and α-glucosidase and corrected diabetes and related complications. In conclusion, the isolated compound morin exhibited antioxidant, antibacterial, antihyperglycemic, and antihyperlipidemic potential in the STZ-induced diabetes rat model. However, additional studies are needed to study their exact mechanism of action.

## 5. Conclusions

The attempt made here was a promising strategy for the use of *E. umbellata* as an alternative medicine in the treatment of diabetes and bacterial infection which most probably may be due to its potentially active morin that was isolated for the first time from the selected plant. The compounds isolated from *E. umbellata* exhibited excellent antioxidant, antibacterial, and antidiabetic effects. Among them, morin displayed a protective effect on STZ-induced diabetes and on injuries to liver and kidneys that normally happen in diabetes and thus could be used as therapeutic candidate for diabetes management and its related complications. Morin also corrected the level of lipid, liver, and kidney parameters, HbA1c level, lipid peroxidation (MDA level), and antioxidant enzymes level. Its protective and regenerative effect on pancreatic β-cells was attributed to activation of β-cells signaling and restoration of histopathological alterations, which led to improved glucose and lipid metabolism. The morin isolated from *E. umbellata* exhibited potent anti-diabetic effects both in vitro and in vivo that could be used in formulation of an alternative medicine for diabetes. However, accurate therapeutic dosages for treating various diseases are yet to be determined. Therefore, random human clinical trials are needed to find safe and effective doses of morin for the treatment of acute and chronic diabetes. Additionally, it is difficult for nutritionists to recommend morin or morin-rich foods at this point, as little is known about their interactions with other foods and about the dose-response and safety profile when consumed with other ingredients or food components. Studies are needed to confirm the mentioned parameters and interactions. Although the antioxidant, antibacterial, and antidiabetic properties of morin against STZ-induced diabetic rats were identified in this study, the molecular mechanism of morin as an antioxidant substance needs to be investigated in future.

## Figures and Tables

**Figure 1 molecules-26-04464-f001:**
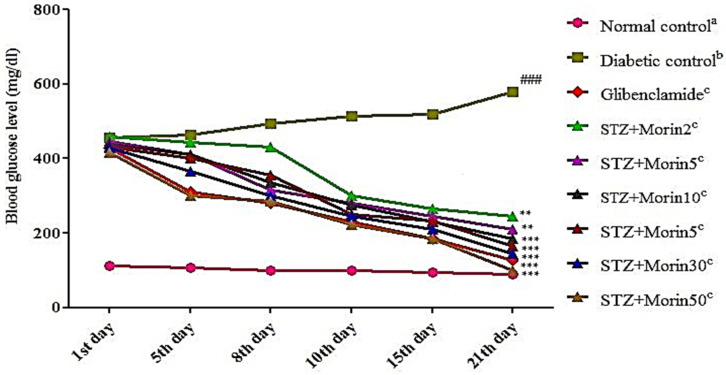
Effect of various treatments on blood glucose level in streptozotocin induced diabetic rats (the values are expressed as mean ± SEM. Each value corresponds to a mean of eight animals per group; ### *p* ˂ 0.001; comparison of normal control^a^ vs. diabetic control^b^ using Student’s *T*-test, ** *p* ˂ 0.01 and *** *p* < 0.001; comparison of diabetic control^b^ vs. glibenclamide^c^ and morin^c^ treated groups using one way ANOVA followed by Dunnett’s post hoc multiple comparison test).

**Figure 2 molecules-26-04464-f002:**
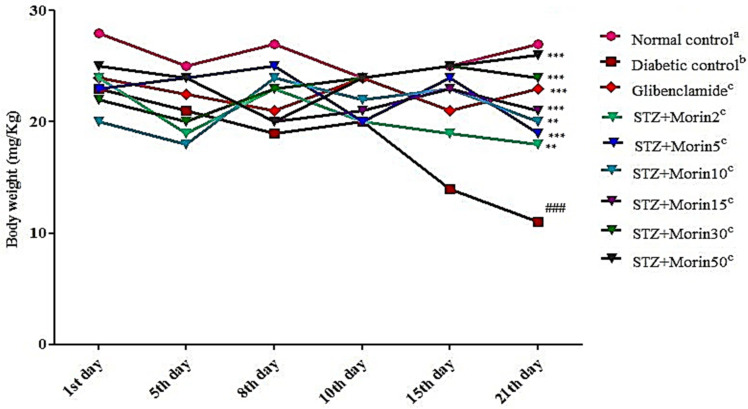
Effects of various treatments on body weight in STZ-induced diabetic rats (the values are expressed as mean ± SEM. Each value corresponds to a mean of eight animals ### *p* ˂ 0.001; comparison of normal control^a^ vs. diabetic control^b^ using Student’s *t-test*, ** *p* ˂ 0.01, *** *p* < 0.001; comparison of diabetic control^b^ vs. glibenclamide^c^ and morin^c^ treated groups using one way ANOVA followed by Dunnett’s post hoc multiple comparison test).

**Figure 3 molecules-26-04464-f003:**
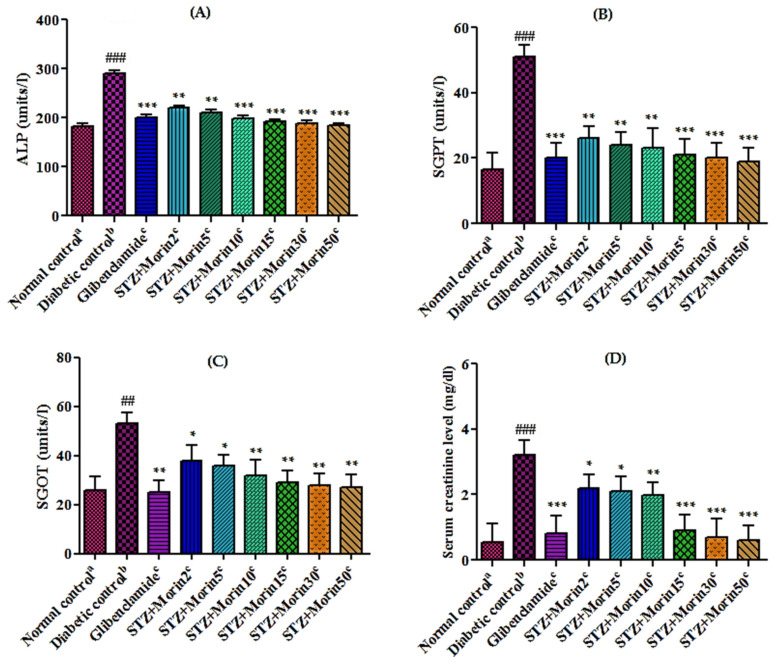
Effects of morin on liver serum biochemical parameters and renal functions in streptozotocin-induced diabetic rats [(**A**) ALP level, (**B**) SGPT level, (**C**) SGOT level, and (**D**) serum creatinine level. The values are expressed as mean ± SEM. Each value corresponds to a mean of eight animals ## *p* ˂ 0.01 and ### *p* ˂ 0.001; comparison of normal control^a^ vs. diabetic control^b^ using Student’s *T*-test, * *p* ˂ 0.05, ** *p* ˂ 0.01, *** *p* < 0.001; comparison of diabetic control^b^ vs. glibenclamide^c^ and morin^c^ treated groups using one way ANOVA followed by Dunnett’s post hoc multiple comparison test].

**Figure 4 molecules-26-04464-f004:**
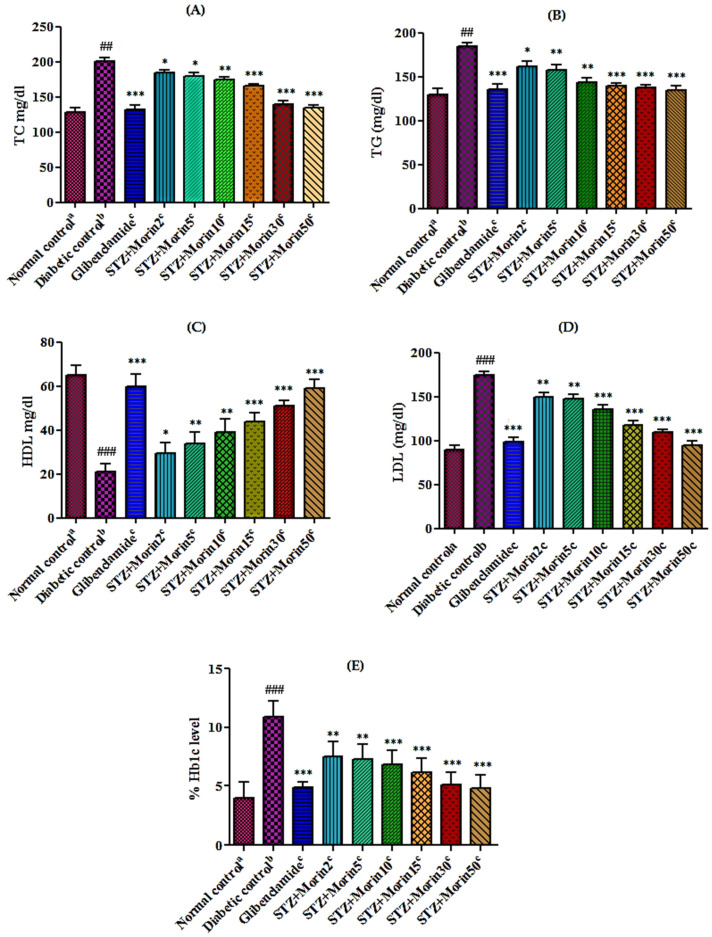
Effect of morin on lipid profile and Hb1c level in streptozotocin induced diabetic rats. [(**A**)TC level, (**B**) TG level, (**C**) HDL level, and (**D**) LDL level, (**E**) Hb1c level. The values are expressed as mean ± SEM. Each value corresponds to a mean of eight animals ## *p* ˂ 0.01; ### *p* ˂ 0.001; comparison of normal control^a^ vs. diabetic control^b^ using Student’s *T*-test, * *p* ˂ 0.05, ** *p* ˂ 0.01, *** *p* < 0.001; comparison of diabetic control^b^ vs. glibenclamide^c^ and morin^c^ treated groups using one way ANOVA followed by Dunnett’s post hoc multiple comparison test].

**Figure 5 molecules-26-04464-f005:**
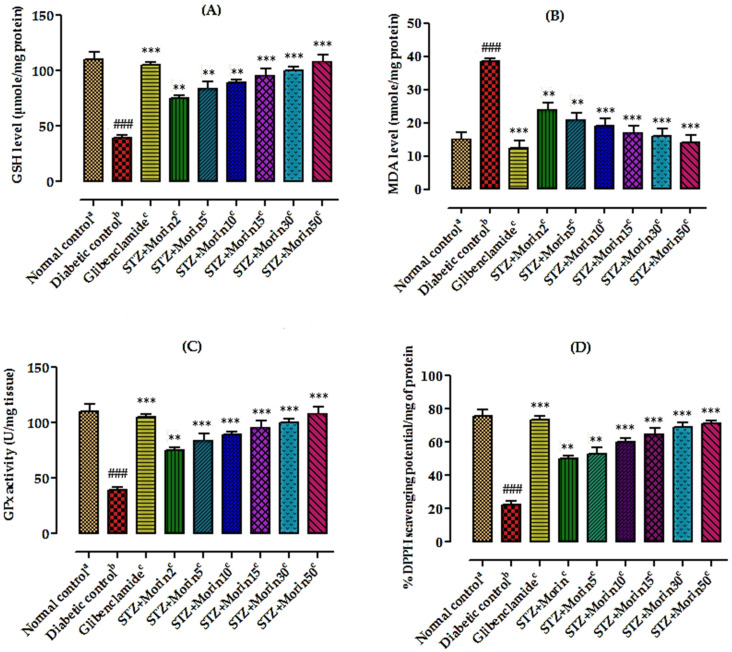
Effects of morin (2, 5, 10, 15, 30, and 50 µg/mL) and glibenclamide (5 mg/kg) on oxidative stress markers in pancreas and intracellular ROS level in STZ-induced diabetic rats. (**A**) Reduced glutathione (GSH), (**B**) MDA level, (**C**) glutathione peroxidase (GPx) level, and (**D**) % DPPH scavenging potential in STZ-induced diabetic rats. The values are expressed as mean ± SEM. Each value corresponds to a mean of eight animals ### *p* ˂ 0.001; comparison of ^a^(normal control) vs. ^b^(diabetic control) using Student’s *T*-test, ** *p* ˂ 0.01, *** *p* < 0.001; comparison of ^b^(diabetic control) vs. ^c^(glibenclamide and morin treated groups) using one way ANOVA followed by Dunnett’s post hoc multiple comparison test.

**Figure 6 molecules-26-04464-f006:**
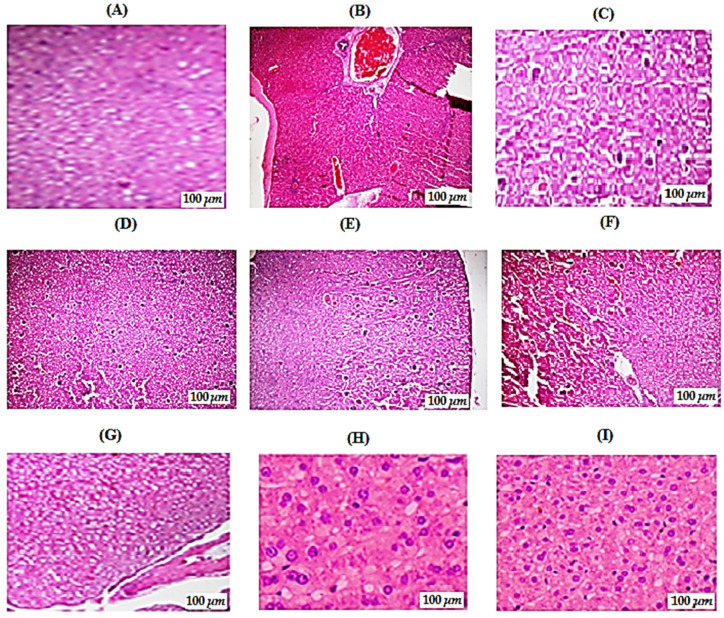
Effects of morin (2, 5, 10, 15, 30, and 50 mg/kg) and glibenclamide (5 mg/kg) on pancreas histopathology in STZ-induced diabetic rats. (**H**,**E**) staining; 40× and 100×, scale bar = 100 μm: (**A**) control group; (**B**) diabetic control group; (**C**) glibenclamide (5 mg/kg/overly) treated group; (**D**,**E**) morin (2, 5 mg/kg/orally) treated group; (**F**,**G**) morin (10, 15 mg/kg/orally) treated groups; (**H**,**I**) morin (30 and 50 mg/kg/orally) treated group.

**Figure 7 molecules-26-04464-f007:**
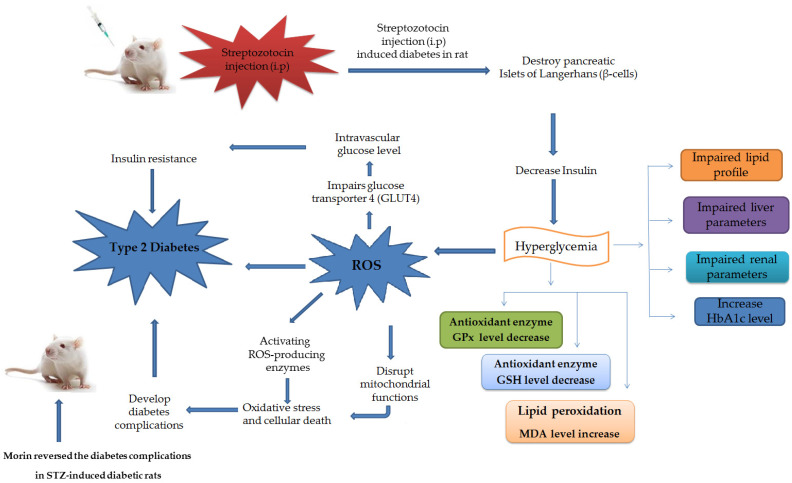
The potential effect of morin on streptozotocin-induced diabetic rat model.

**Table 1 molecules-26-04464-t001:** Experimental design and various treated groups used in in vivo study.

Group	Group Category	Treatment Given	Route
Group I	Normal control	Normal saline 8 mL/kg	p.o.
Group II	Diabetic control	STZ (50 mg/kg) + normal saline	i.p.
Group III	Positive control	STZ (50 mg/kg) + glibenclamide (5 mg/kg)	i.p./p.o.
Group IV	Compound treated group	STZ (50 mg/kg) + morin (2 mg/kg)	i.p./p.o.
Group V	Compound treated group	STZ (50 mg/kg) + morin (5 mg/kg)	i.p/p.o.
Group VI	Compound treated group	STZ (50 mg/kg) + morin (10 mg/kg)	i.p./p.o.
Group VII	Compound treated group	STZ (50 mg/kg) + morin (15 mg/kg)	i.p./p.o.
Group VIII	Compound treated group	STZ (50 mg/kg) + morin (30 mg/kg)	i.p./p.o.
Group IX	Compound treated group	STZ (50 mg/kg) + morin (50 mg/kg)	i.p./p.o.

STZ; streptozotocin, p.o.; orally, i.p.; intraperitoneal.

**Table 2 molecules-26-04464-t002:** Percent DPPH and ABTS free radical scavenging activity of compounds (morin, phloroglucinol, 1-hexyl benzene) isolated from *Elaeagnus umbellata* berries at various concentrations.

S. No	Sample	Concentration (µg/mL)	% DPPH Scavenging	DPPH IC_50_ (µg/mL)	% ABTS Scavenging	ABTS IC_50_ (µg/mL)
			Mean ± SEM		Mean ± SEM	
1	Morin	1000	88 ± 1.1 ^ns^	37	88 ± 1.0 ^ns^	40
		500	83 ± 0.5 ^ns^		80 ± 1.0 ^ns^	
		250	76 ± 1.0 *		75 ± 0.5 ^ns^	
		125	68 ± 1.2 ***		64 ± 1.2 **	
		62.5	59 ± 1.3 ***		57 ± 1.4 ***	
		31.05	54 ± 0.5 ***		45 ± 0.5 ***	
2	Phloroglucinol	1000	67 ± 0.4 ***	105	64 ± 1.1 ***	112
		500	57 ± 1.0 ***		51 ± 0.5 ***	
		250	51 ± 1.0 ***		45 ± 1.0 ***	
		125	40 ± 0.5 ***		39 ± 1.2 ***	
		62.5	36 ± 1.1 ***		33 ± 0.5 ***	
		31.05	32 ± 0.5 ***		25 ± 1.0 ***	
3	1-Hexyl benzene	1000	59 ± 1.5 ***	118	55 ± 1.1 ***	129
		500	53 ± 2.1 ***		47 ± 2.2 ***	
		250	46 ± 0.5 ***		39 ± 1.5 ***	
		125	40 ± 1.1 ***		31 ± 1.0 ***	
		62.5	35 ± 0.4 ***		26 ± 0.5 ***	
		31.05	28 ± 1.1 ***		19 ± 1.4 ***	
4	Ascorbic acid	1000	91 ± 0.5	29	90 ± 0.5	32
		500	86 ± 1.0		84 ± 1.1	
		250	80 ± 0.4		78 ± 0.5	
		125	77 ± 0.4		70 ± 0.5	
		62.5	73 ± 0.5		64 ± 1.0	
		31.05	69 ± 0.4		56 ± 1.1	

Data are expressed as mean ± SEM, ^ns^ = not significant, * *p* ˂ 0.05, ** *p* ˂ 0.01, *** *p* ˂ 0.001; comparison of morin vs. positive control ascorbic acid using Student’s T-test. The half maximal inhibitory concentration (IC_50_) is that concentration of a compound capable of scavenging 50% of the free radicals (DPPH and ABTS).

**Table 3 molecules-26-04464-t003:** Percent α-glucosidase and α-amylase inhibition potential of compounds (morin, phloroglucinol, 1-hexyl benzene) isolated from of *Elaeagnus umbellata* berries at various concentrations.

Sample	Concentration (µg/mL)	% α-Amylase Inhibition	α-Amylase IC_50_ (µg/mL)	% α-Glucosidase Inhibition	**α-Glucosidase IC_50_ (µg/mL)**
		Mean ± SEM		Mean ± SEM	
Morin	1000	75 ± 1.5 ***	55	79 ± 1.0 **	40
	500	67 ± 1.4 ***		66 ± 0.4 ***	
	250	60 ± 0.5 **		59 ± 1.4 **	
	125	54 ± 1.4 **		52 ± 0.5 ***	
	62.5	44 ± 2.1 ***		48 ± 0.3 ***	
	31.05	37 ± 1.1 **		42 ± 1.4 ***	
Phloroglucinol	1000	63 ± 0.5 ***	95	55 ± 2.1 ***	110
	500	55 ± 1.1 ***		48 ± 0.5 ***	
	250	42 ± 0.4 ***		43 ± 0.4 ***	
	125	39 ± 1.5 ***		37 ± 1.5 ***	
	62.5	32 ± 1.4 ***		30 ± 1.4 ***	
	31.05	29 ± 1.0 ***		27 ± 1.0 ***	
1-Hexyl benzene	1000	55 ± 1.5 ***	115	58 ± 2.1 ***	105
	500	49 ± 2.1 ***		50 ± 1.2 ***	
	250	44 ± 0.5 ***		45 ± 0.5 ***	
	125	37 ± 1.4 ***		39 ± 1.1 ***	
	62.5	28 ± 2.1 ***		32 ± 0.4 ***	
	31.05	22 ± 1.1 ***		28 ± 1.2 ***	
Standard acarbose	1000	90 ± 0.5	25	89 ± 0.5	28
	500	83 ± 0.5		82 ± 0.4	
	250	75 ± 1.1		77 ± 1.0	
	125	69 ± 0.5		71 ± 0.5	
	62.5	62 ± 0.5		68 ± 0.5	
	31.05	48 ± 0.4		60 ± 0.4	

Data are expressed as mean ± SEM, ** *p* ˂ 0.01, *** *p* ˂ 0.001; comparison of morin vs. positive control acarbose using Student’s *T*-test. The half maximal inhibitory concentration (IC_50_) is that concentration of a compound capable of inhibiting 50% of the carbohydrate digesting enzymes (α-glucosidase and α-amylase).

**Table 4 molecules-26-04464-t004:** Zone of inhibitions of the morin against various bacterial strains.

S. No	Bacterial Strains	Zone of Inhibition (mm)			
		Morin (µg/mL)	Phloroglucinol (µg/mL)	1-Hexyl Benzene (µg/mL)	Imipenem (µg/mL) (Positive)
1	*Escherichia coli*	19 ± 1.1 **	15 ± 0.3 ***	9.3 ± 1.1 ***	33 ± 0.2
2	*Bacillus cereus*	17 ± 0.1 **	13 ± 1.2 ***	8.5 ± 0.1 ***	30 ± 0.5
3	*Salmonella typhi*	16 ± 1.2 ***	12 ± 1.1 ***	7.4 ± 1.2 ***	33 ± 0.1
4	*Klebsiella pneumonia*	15 ± 0.5 ***	11 ± 0.5 ***	6.3 ± 0.5 ***	29 ± 0.5
5	*Pseudomonas aeruginosa*	14 ± 0.5 ***	10 ± 1.1 ***	5.5 ± 0.5 ***	27 ± 1.1
6	*Proteus mirabilis*	11 ± 1.5 ***	7.5 ± 0.2 ***	4.5 ± 1.5 ***	30 ± 0.5
7	*Staphylococcus aureus*	13 ± 0.4 ***	9.5 ± 2.1 ***	3.4 ± 0.4 ***	25 ± 0.3

Values significantly different as compared to standard drug i.e., ***: *p* < 0.001, and **: *p* < 0.01.

**Table 5 molecules-26-04464-t005:** Minimum inhibitory concentrations of morin against Gram-positive and Gram-negative bacterial strains.

S. No	Bacterial Strains	Minimum Inhibitory Concentrations (µg/mL)			
		Morin	Phloroglucinol	1-Hexyl Benzene	Ampicillin
1	*Escherichia coli*	25 ± 1.1 ***	80 ± 1.4 ***	95 ± 2.1 ***	1.0 ± 0.1
2	*Bacillus cereus*	21 ± 1.2 ***	105 ± 1.6 ***	135 ± 1.5 ***	1.6 ± 0.0
3	*Salmonella typhi*	28 ± 1.0 ***	175 ± 2.0 ***	112 ± 1.6 ***	1.8 ± 0.1
4	*Klebsiella pneumonia*	30 ± 1.4 ***	166 ± 1.5 ***	178 ± 2.2 ***	1.9 ± 1.2
5	*Pseudomonas aeruginosa*	44 ± 1.5 ***	180 ± 2.1 ***	200 ± 1.5 ***	1.2 ± 0.5
6	*Proteus mirabilis*	55 ± 1.4 ***	205 ± 1.5 ***	250 ± 1.2 ***	1.9 ± 1.1
7	*Staphylococcus aureus*	68 ± 1.3 ***	192 ± 2.3 ***	205 ± 2.1 ***	1.4 ± 0.5

Values significantly different as compared to standard drug i.e., ***: *p* < 0.001.

## Data Availability

The data presented in this manuscript belong to the PhD research work of Mrs. Nausheen Nazir and has not been deposited in any repository yet. However, the data are available to the researchers upon request.

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
