# Peer review of "Antioxidants Isolated from Elaeagnus umbellata (Thunb.) Protect against Bacterial Infections and Diabetes in Streptozotocin-Induced Diabetic Rat Model"

_molecules, 2021, doi:10.3390/molecules26154464_

Round 1

Reviewer 1 Report

The authors fulfilled my requests, so I suggest the acceptance of their revised MS.

Author Response

Reviewer 1:

Comments and Suggestions for Authors

The authors fulfilled my requests, so I suggest the acceptance of their revised MS.

  • Answer: Thank you so much worthy reviewer for acceptance of our revised manuscript.

Reviewer 2 Report

Reviewer comments

TITLE

Add plant species name

Add geographical area (i.e. Pakistan)

ABSTRACT

Start with opening problem statement sentence (what led to the need for this investigation)

No results represented on phloroglucinol or 1-hexyl benzene isolation

Selected bacterial strains not indicated

Tested dosages not indicated

Over presentation of morin results at the costs of other equally important data – rebalance

“Morin exhibited highest biological potentials …” as opposed to?

Keywords

Do not use keywords already captured in title of paper.

INTRODUCTION

Overly long paragraphs lead to loss of focus and identity of individual paragraphs.

Line 48: ROS however, …” – always add comma before and after however.

Line 51: First indicate why is ROS over-produced, before indicating its impact

Line 58: “Clinical diabetes …” start of new paragraph

Line 63: 2003 data too old (18 years ago) to be of value

Line 87 vs 90: Researchers have conducted (line 8) vs [7] – are you citing a secondary source? – incorrect citation format

Line 113: family names not typeset in italics

Line 120: cannot cite source 112.

Line 125: “Among the compounds…” start of new paragraph

Line 126: Berries fruit – a berry is per definition a fruit.  Just use berry – correct throughout paper.

Line 142: “Therefore, many plant …” start of new paragraph

Line 142: “… plant species…”

Line 150: Rephrase “to combat such infections the in use currently employed …”

Line 158-160: Is this a result stemming from this paper?

MATERIALS AND METHODS

Line 180: a local area

Line 378/9: Rephrase “…Experiments) guidelines [29], morin …”

Line 451: Rephrase “…to the reported assay [36] the stained slides…”

RESULTS

Line 466: rephrase: “… (Table 2). While Phloroglucinol …”

Line 469: check spacing between % and antiradical

Line 471: check spacing between 32 and mg/Ml

Line 519” Rephrase “... hence the plant species …”

Line 572: Rephrase “… weight showed resulted in significant …”

Line 599-600: Nonsensical sentence – delete

Line 653/4 & 657/8: diameter of Islets of Langerhans cells (µm?)

Line 667: rephrase “… Langerhans was declined …”

Line 670: define little bit restoration

Line 673: define negligible

Line 680: rephrase “…50mg/kg/orally has shown …” to “…50mg/kg/orally showed …”

DISCUSSION

Lines 688-702: remove – no need for another introduction paragraph and duplication of information presented in Introduction section – get straight into discussing first results

Line 736/7: reported literatures” – such as?

Line 770: rephrase “Histopathological study results …”

Line 774: rephrase “reported literature [49] …”

CONCLUSION

 = Not a summary, which domain of abstract. Replace with:

  • Indicate what is new and innovative in the investigation.
  • Indicate the gaps in this investigation and how to close them.
  • Indicate the holistic conclusion.
  • Indicate the holistic recommendations.
  • Indicate the real-world applications of the data presented.

Author Response

Reviewer 2:

Comments and Suggestions for Authors

Dear Editor/ reviewer,

Thank you very much for kind review and comments concerning our manuscript. We appreciate the hard work of reviewers as they fairly pointed out errors and mistakes in our manuscript. Once again thank you for your informative suggestion related the whole manuscript, which makes the manuscript more valuable for the readers. We have tried our best and revise the manuscript in line with comments raised by reviewer. Corrections made have been highlighted as Blue.

Please find below the point by point responses to the reviewer’s comments and suggestions.

Reviewer comments

TITLE

Add plant species name

  • Answer: Worthy reviewer the plant species name has been added in the revised title.

Add geographical area (i.e. Pakistan)

  • Answer: Worthy reviewer geographical area (i.e. Pakistan) has already been added in the material and method section. As we are working with our research team in Saudi Arabia, if we add such things in title, it become objectionable for them. Hope worthy reviewer, will have understand our problem

ABSTRACT

Start with opening problem statement sentence (what led to the need for this investigation)

  • Answer: Worthy reviewer the required statement has been added accordingly.

No results represented on phloroglucinol or 1-hexyl benzene isolation

  • Answer: The details related phloroglucinol or 1-hexyl benzene isolation has been added accordingly.

Selected bacterial strains not indicated

  • Answer: Worthy reviewer the tested strains has been added in the required section.
  • Escherichia coli, Bacillus cereus, Salmonella typhi, Klebsiella pneumonia, Pseudomonas aeruginosa, and Proteus mirabilis.

Tested dosages not indicated

  • Answer: Tested doses of morin have been added in the revised suggested section.

Over presentation of morin results at the costs of other equally important data – rebalance

  • Answer: Worthy reviewer, based on strong in vitro results (antioxidants and antidiabetic) as exhibited by morin, the in vivo antidiabetic study was designed only on morin that’s why most of the results section composed of morin only.

“Morin exhibited highest biological potentials …” as opposed to?

  • Answer: Corrected accordingly.

Keywords

Do not use keywords already captured in title of paper.

  • Answer: Corrected accordingly.

INTRODUCTION

Overly long paragraphs lead to loss of focus and identity of individual paragraphs.

  • Answer: Worthy reviewer the paragraphs size has been reduce and extra information has been removed accordingly.

Line 48: ROS however, …” – always add comma before and after however.

  • Answer: Corrected accordingly.

Line 51: First indicate why is ROS over-produced, before indicating its impact

  • Answer: Worthy reviewer paragraph related ROS production has been added and the whole paragraph has been revised.

Line 58: “Clinical diabetes …” start of new paragraph

  • Answer: Corrected accordingly.

Line 63: 2003 data too old (18 years ago) to be of value

  • Answer: Old data has been removed in the revised manuscript.

Line 87 vs 90: Researchers have conducted (line 8) vs [7] – are you citing a secondary source? – incorrect citation format

  • Answer: Worthy reviewer the citation [7] has been corrected in the revised paragraph, which in the revised manuscript becomes [8], due to addition of some data in the start of introduction section.

Line 113: family names not typeset in italics

  • Answer: Corrected accordingly.

Line 120: cannot cite source 112.

  • Answer: Worthy reviewer it was mistakenly written as 112, this citation is 12, which is corrected accordingly.

Line 125: “Among the compounds…” start of new paragraph

  • Answer: Corrected.

Line 126: Berries fruit – a berry is per definition a fruit.  Just use berry – correct throughout paper.

  • Answer: The worthy reviewer suggestion has been honoured and use only beery in the whole manuscript.

Line 142: “Therefore, many plant …” start of new paragraph

  • Answer: Corrected accordingly.

Line 142: “… plant species…”

  • Answer: Corrected.

Line 150: Rephrase “to combat such infections the in use currently employed …”

  • Answer: The whole paragraph has been revised, as the concept of recommended statement has already been given in this paragraph: ‘’The overwhelming resistance developed by microbes to available drugs …………………….’’.

Line 158-160: Is this a result stemming from this paper?

  • Answer: Yes worthy reviewer, however, the paragraph has been revised to remove the confusion.

MATERIALS AND METHODS

Line 180: a local area

  • Answer: Corrected accordingly.

Line 378/9: Rephrase “…Experiments) guidelines [29], morin …”

  • Answer: Corrected accordingly.

Line 451: Rephrase “…to the reported assay [36] the stained slides…”

  • Answer: Corrected accordingly.

RESULTS

Line 466: rephrase: “… (Table 2). While Phloroglucinol …”

  • Answer: Corrected accordingly.

Line 469: check spacing between % and antiradical

  • Answer: Corrected accordingly in whole manuscript.

Line 471: check spacing between 32 and mg/Ml

  • Answer: Corrected accordingly.

Line 519” Rephrase “... hence the plant species …”

  • Answer: Corrected.

Line 572: Rephrase “… weight showed resulted in significant …”

  • Answer: Corrected accordingly.

Line 599-600: Nonsensical sentence – delete

  • Answer: Worthy reviewer the suggested sentence has been deleted accordingly.

Line 653/4 & 657/8: diameter of Islets of Langerhans cells (µm?)

  • Answer: Worthy reviewer the Line 653/4 & 657/8 has been modified and confusion in diameter has been removed.

Line 667: rephrase “… Langerhans was declined …”

  • Answer: Corrected.

Line 670: define little bit restoration

  • Answer: The suggested paragraph has been revised.

Line 673: define negligible

  • Answer: The suggested paragraph has been revised.

Line 680: rephrase “…50mg/kg/orally has shown …” to “…50mg/kg/orally showed …”

  • Answer: Corrected.

DISCUSSION

Lines 688-702: remove – no need for another introduction paragraph and duplication of information presented in Introduction section – get straight into discussing first results

  • Answer: Removed accordingly.

Line 736/7: reported literatures” – such as?

  • Answer: Corrected accordingly.

Line 770: rephrase “Histopathological study results …”

  • Answer: Corrected accordingly.

Line 774: rephrase “reported literature [49] …”

  • Answer: Corrected accordingly.

CONCLUSION

 = Not a summary, which domain of abstract. Replace with:

Indicate what is new and innovative in the investigation.

Indicate the gaps in this investigation and how to close them.

Indicate the holistic conclusion.

Indicate the holistic recommendations.

Indicate the real-world applications of the data presented.

Answer: Worthy reviewer thank you for your informative suggestions related conclusion. All the points you mentioned has been added in the revised conclusion. 

This manuscript is a resubmission of an earlier submission. The following is a list of the peer review reports and author responses from that submission.

Round 1

Reviewer 1 Report

This work is quite similar to your previous work Nazir, N.; Zahoor, M.; Ullah, R.; Ezzeldin, E.; Mostafa, GAE. Curative Effect of Catechin Isolated from Elaeagnus Umbellata Thunb. Berries for Diabetes and Related Complications in Streptozotocin-Induced Diabetic Rats Model. Molecules. 2021, 26(1), 137. 

Reviewer 2 Report

General comments

 This study by Nazir et al. investigates the anti-bacterial, anti-oxidant, and anti-diabetic activity effects of morin, phloroglucinol, 1-hexyl benzene isolated from the berries of Elaeagnus umbellata. Among these compounds, morin showed the most effective anti- the anti-bacterial, anti-oxidant, and anti-diabetic properties. It seems that the study was carefully designed, performed, described and a variety of in vivo and in vitro methods were used. However, several minor issues need to be addressed.

Minor comments

  1. The Introduction part is disproportionally long. The author should be more focused and summarize the most important background information on 1 page.
  2. It would be better to present the Results of Tables 6 and 7 on line charts and Tables 8 and 9 on bar graphs.

Reviewer 3 Report

The manuscript  entitled “Antioxidants protect against bacterial infections and diabetes in streptozotocin-induced diabetic rats model” have demonstrated antioxidant, antibacterial and antidiabetic potential of morin isolated from E.umbellata berries. Authors characterized isolated compounds, its antioxidants capacity (DPPH and ABTS scavenging activity) and antimicrobial activity against Gram-positive and Gram-negative bacteria  and in vivo studies (glucose level, lipid profile and peroxidation, antioxidants enzymes and oxidative stress markers). The mannscript is very interesting and the experiments are generally well planned and described. However, Authors should correct manuscript according to the suggestion.

Minor issues:

In all text please correct name of „S. aurous” to S.aureus

Materials and methods

2.10 and 2.11”: please give more details (inoculum preparation and size). Comparison between antimicrobial effects of phytochemicals is difficult, because of the use of different criteria (inoculum preparation and size, growth medium, incubation conditions and endpoints determination). The most recognized standards are provided by the CLSI and the European Committee on Antimicrobial Susceptibility Testing (EUCAST). This standards allow researchers to compare results. Morover, Authors should provide range of phytochemicals concentration for MICs estimation. authors only give one concentration 100 µg/mL.

Results

Table 2.  Please provide IC50 defintion and correct IC50 values. Based on information (scavenging % of DPPH and ABTS and range of phytochemicals concentration), in my opinion IC50 values are incorrect.